# N-acetyltransferase Gene Variants Involved in Pediatric Idiosyncratic Drug-Induced Liver Injury

**DOI:** 10.3390/biomedicines12061288

**Published:** 2024-06-11

**Authors:** María Luisa Alés-Palmer, Francisco Andújar-Vera, Iván Iglesias-Baena, Paloma Muñoz-de-Rueda, Esther Ocete-Hita

**Affiliations:** 1Department of Pediatrics, University of Granada, 18016 Granada, Spain; estherocete@ugr.es; 2Department of Pediatrics, “Virgen de las Nieves” University Hospital, 18014 Granada, Spain; 3Bioinformatic Unit, Instituto de Investigación Biosanitaria ibs.GRANADA, 18012 Granada, Spain; fandujar@ibsgranada.es; 4Research Unit, GenActive Clinic & Research, 18193 Granada, Spain; genactive.info@gmail.com; 5Research Support Unit, Instituto de Investigación Biosanitaria ibs.GRANADA, 18012 Granada, Spain; palomalancha@ibisgranada.es; 6Instituto de Investigación Biosanitaria ibs.GRANADA, 18012 Granada, Spain

**Keywords:** DILI, whole-exome sequencing, risk factors, hepatotoxicity, liver injury, children, NAT-2 polymorphism

## Abstract

Idiosyncratic drug-induced liver injury (DILI) is a complex multifactorial disease in which the toxic potential of the drug, together with genetic and acquired factors and deficiencies in adaptive processes, which limit the extent of damage, may determine susceptibility and make individuals unique in their development of hepatotoxicity. In our study, we sequenced the exomes of 43 pediatric patients diagnosed with DILI to identify important gene variations associated with this pathology. The result showed the presence of two variations in the NAT2 gene: c.590G>A (p.Arg197Gln) and c.341T>C (p.Ile114Thr). These variations could be found separately or together in 41 of the 43 patients studied. The presence of these variations as a risk factor for DILI could confirm the importance of the acetylation pathway in drug metabolism.

## 1. Introduction

Drug-induced liver injury (DILI) is a rare pathology (14 to 19 cases per 100,000 people) causing 40 to 50% of acute liver failure in adults [1]. This problem has been less studied in the pediatric population despite the fact that in more than 40% of liver transplants, the reason for the fulminant liver failure that could be related to this health problem is unknown [1,2]. It is usually caused by a variety of drugs, supplements and herbal products. DILI is a growing problem that even affects the manufacture of drugs, including others that can be found on the market. In children, this problem is increased by the fact that most drugs are used without an indication in the technical data sheet or even with an indication different from the one included in the technical data sheet due to the difficulty of conducting clinical trials in the child population [2,3]. DILI can be divided into intrinsic reactions (which are dose-dependent and therefore predictable once a specific threshold concentration has been consumed) and idiosyncratic reactions, which are not dose-dependent and therefore cannot be explained by the known pharmacological properties of the drug [3,4]. Since the end of the Human Genome Project, genetic analysis techniques have grown at an exponential rate. Whole-exome sequencing (WES) is one of the most efficient techniques for high-throughput genomic analysis, sequencing more than 90% of an individual’s coding DNA. Compared with whole-genome sequencing, the exome represents approximately 1% of the total genome, allowing an improvement in both effort and cost-effectiveness in mutation identification. This is because many disease-causing mutations are located in protein-coding regions [5,6]. On the other hand, pathological genetic aberrations in regulatory regions or poorly sequenced exome regions will remain unidentified. Despite these limitations, WES has proven successful in the identification of both new genes and disease pathways, appearing as the main technique in many scientific articles [7].

The relationships between human variations and phenotypes were queried using the ClinVar database (https://www.ncbi.nlm.nih.gov/clinvar/ (accessed on 2 May 2024)), which provides public and open access to its reports.

We hypothesize an improved understanding of the molecular mechanisms underlying the development and progression of DILI with the support of sequenced exome analysis. 

Current high-throughput sequencing technologies are capable of uncovering genomic sequences whose analysis can be used to characterize genomic alterations (common or rare) in the development of DILI. The exome study approach may eventually provide important biological data supported by the genetic alterations suffered by DILI patients. The molecular target discovery of novel proteins could be used for targeted therapies or the identification of networks of genetic alterations relevant to the clinical context of the pathology of this study [8].

N-acetyltransferase (NAT2) is a cytosolic phase II conjugation enzyme that is encoded by the NAT2 gene present on chromosome 8. It is a key human enzyme that plays a role in the detoxification and removal of many drugs, including carcinogenic arylamine compounds [9].

NAT2 is one of only 2 N-acetyltransferase genes in humans; the other, NAT1, shows little variation between individuals, whereas NAT2 is known to have over 23 variants. N-acetyltransferases are enzymes acting primarily in the liver to detoxify a large number of chemicals and drugs. The NAT2 acetylation polymorphism is important because of its primary role in the activation and/or deactivation of many chemicals in the body’s environment. In turn, this can affect an individual’s cancer risk [10].

Individuals can be classified as either rapid or slow metabolizers (i.e., detoxifiers). In general, slow metabolizers have higher rates of certain types of cancer and are more susceptible to adverse effects from chemicals metabolized by NAT2 [11,12,13].

It takes two slow-metabolizer alleles to give rise to a slow-metabolizer phenotype, or to put it another way, the rapid-metabolizer allele is dominant over the slow metabolizer; therefore, only one is needed to be a rapid metabolizer.

The NAT2 gene is highly polymorphic, with more than a hundred variant alleles. Genetic variation in the NAT2 gene plays an important role in isoniazid-induced hepatotoxicity [14]. Most of the variants (SNPs) are located within the 873 bp intron-free coding region of the NAT2 gene. Among the different alleles, four variant alleles—341 T>C (rs1801280; NAT2*5), 590 G>A (rs1799930; NAT2*6), 857 G>A (rs1799931; NAT2*7) and 191 G>A (rs1801279; NAT2*14)—are studied mainly because of their influence on the acetylation activity of the NAT2 enzyme [15].

To enhance our understanding of the relationship between genetic polymorphisms and susceptibility to drug-induced liver injuries in the pediatric population, this study focuses on specific polymorphisms of the NAT2 gene. Previous studies have shown the importance of genetic variations in pharmacokinetics and drug response, but there is a significant gap in the literature regarding how these variations affect children, who may have distinct metabolic profiles compared to adults.

Therefore, the aim of this study was to improve our understanding of the development of pediatric DILI by determining drug-responsiveness, pathogenicity or alterations of uncertain significance in order to identify new strategies for therapeutic approaches and personalized therapy.

## 2. Materials and Methods

### 2.1. Population Study

The population of this study included 43 pediatric patients between the ages of 0 and 15 years who were selected by physicians from participating hospitals and had liver disease associated with the ingestion of medications or herbal products. The operational structure of the registry, the data recording process and the characterization of cases are described in detail in Ocete Hita E (2013) [16]. Diagnoses of causality were made using the CIOMS/RUCAM evaluation method [17]. The characteristics of the pediatric patients included in the study are summarized in Appendix A.

All these patients were enrolled in the Spanish registry of hepatotoxic reactions in the pediatric population. The samples were collected and managed by the Biobank of the Public Health System of Andalusia in accordance with its internal procedures.

Due to the difficulty in obtaining healthy pediatric samples and to ensure that they were healthy participants with no history of liver disease, we obtained samples from 28 adults as controls to be compared by WES with our pediatric samples.

### 2.2. Ethics Approval and Consent to Participate

This study was conducted in accordance with the Declaration of Helsinki (as revised in 2013). Ethical approval for the study was obtained from the Granada Provincial Research Ethics Committee (ethics permission number: 0057-M1-20). Written informed consent to participate was obtained from the parents or legal guardians of all participants (under the age of 18) or from participants (for control individuals).

### 2.3. Sample Processing

DNA was isolated from peripheral blood samples obtained from all patients. The commercial HigherPurity™ Blood Genomic DNA Extraction Kit (Canvax, Biotech, Córdoba, Spain) was used for DNA purification according to the manufacturer’s instructions and was used to prepare fragment libraries suitable for massively parallel paired-end sequencing.

The quality and quantity of the gDNA were tested using 3 µL per sample using a 2200 TapeStation system. (A) The amount of DNA was measured by the picogreen method (Invitrogen^TM^, cat. #P7589, Waltham, MA, USA) using Victor 3 fluorometry. (B) DNA status: this was performed using the gel electrophoresis method because this method is a powerful means of revealing the condition (including the presence or absence) of DNA in a sample.

### 2.4. Whole-Exome Sequencing (WES)

A portion of the DNA obtained (50 ng) was fragmented, processed and amplified to generate a sequencing library using the Human Core Exome design and protocol from Twist Bioscience. Twist NGS Target Enrichment is a system based on 130-mer biotinylated ultralarge biotinylated cDNA bait to capture regions of interest from the NGS genomic fragment library and enrich them.

Quality control of the libraries was then performed; first, the size of the PCR-enriched fragments that make up the libraries was checked using an Agilent Technologies Bioanalyzer 2100 with a DNA 1000 chip. Secondly, the quantity of the libraries was checked by qPCR according to the Illumina qPCR quantification protocol guide (KAPA Library Quantification kits for Illumina Sequencing platforms).

Sequencing of the libraries was carried out on an Illumina NovaSeq 6000 Sequencing System. The primary calculations of the data obtained were performed in the integrated RTA (Real-Time Analysis) software v3.4.4. The processed files were transformed to FASTQ using the Illumina bcl2fastq v2.20.0 package, generating a set of paired reads with a size of about 150 bases in length for all samples.

### 2.5. WES Data Analysis

The FASTQ files obtained in the previous section were quality checked by FASTQC and, subsequently, the paired reads were mapped against the human revertant genome (version GRCh38.p13) using the BWA-MEM alignment program (bwa-0.7.17), thus generating .bam files of aligned reads without disordered sequences and alternate haplotypes. Possible duplicate (identical) reads due to PCR amplification were removed using the MarkDuplicates program of the Picard-tools analysis package (v2.18.2). Reads with identical starting positions were considered duplicates and were reduced to a single read.

The .bam files were recalibrated with the Base Quality Score Recalibration (BQSR) program, which uses machine learning algorithms to model sequencing errors empirically and adjust quality scores.

The recalibrated .bam files were used to identify changes (variants) with respect to the reference sequence using GATK’s Haplotype Caller algorithm (v4.0.5.1), thus generating .vcf files.

The identified variants were filtered with GATK’s VariantFiltration tool, followed by functional annotation of the variants using SnpEff (v5.0e 2021-03-09) and filtered with dbSNP (v154) and SNPs from the 1000 genomes project (Phase3). The final file format was vcf. Then, an in-house program and SnpEff was used to annotate variants with additional databases, including ESP6500 (ESP6500SI_V2), ClinVar, dbNSFP (v4.2c) and ACMG.

### 2.6. Variant Selection

The set of variants identified by whole-exome sequencing was narrowed down according to the phenotype described by ClinVar. For this purpose, those variations that were pathogenic or probably pathogenic, responded to drugs or were variables of uncertain significance were selected. In addition, gene variants with a synonymous effect were discarded.

Enrichment analysis of the proteins involved was performed using SRING v12.0.

### 2.7. Validation of DILI-Related Variants and Frequency in Our Samples

The variants found to be relevant would be validated by linkage of DILI-associated genes. Two databases were used to extract information about these genes involved in DILI. DisGeNET (https://www.disgenet.org/ (accessed on 12 May 2023)) (GDA_score > 0.3) was used on the one hand, and Phenopedia (https://phgkb.cdc.gov/ (accessed on 12 May 2023)) on the other. The genes obtained in both databases were compared with the genes of the variants resulting from the previous section.

To determine the significance of these mutations in our samples, we discarded those that did not occur in at least 80% of our samples, i.e., each gene should contain at least one variation in 35 of the 43 samples that were sequenced.

### 2.8. Predicting the Functional Effect of Amino Acid Substitutions

Once variants with a high frequency in our samples that were related to DILI were obtained, computational tools such as SIFT (https://www.ncbi.nlm.nih.gov/pmc/articles/PMC3394338/ (accessed on 12 May 2023)), PROVEAN (https://www.ncbi.nlm.nih.gov/pmc/articles/PMC4528627/ (accessed on 12 May 2023)), Aling GVGD (https://www.ncbi.nlm.nih.gov/pmc/articles/PMC2563222/ (accessed on 25 April 2024)), Panther (https://pubmed.ncbi.nlm.nih.gov/27193693/ (accessed on 25 April 2024)) and PolyPhen 2 (https://pubmed.ncbi.nlm.nih.gov/23315928/ (accessed on 25 April 2024)) were used to determine which of these variants might be causative of the pathology or characteristic phenotype in DILI. Default thresholds were used to determine whether it was a deleterious/pathogenic mutation based on the prediction of each tool.

### 2.9. Statistical Analysis

Categorical variables were summarized using frequencies (%), and group comparisons were conducted using chi-square tests or Fisher’s exact test, when appropriate. A *p*-value < 0.05 was considered significant. The analysis was conducted using R-studio statistical software version 4.2.3.

### 2.10. Genotyping of SNPs in NAT2 by Real-Time qPCR

All genotyping experiments were conducted at the Integrated DNA Technologies (IDT) laboratory (Redwood City, CA, USA). The controls and patient samples were genotyped by rhAmp technology with designed primers for rs1801280 and rs1799930 following the manufacture protocol, and measurements were carried out with a BioRad real time thermocycler. The oligonucleotides were as follows: for rs1801280, Allele Primer 1/rhAmp-F/AATGTAATTCCTGCCGTCAArUGGTC/GT4/, Allele Primer 2/rhAmp-Y/AATGTAATTCCTGCCGTCAGrUGGTC/GT4/ and Locus Primer GCTACATCCCTCCAGTTAACAAATACrAGCAC/GT1/; for rs1799930, Allele Primer 1/rhAmp-F/TACTTATTTACGCTTGAACCTCGrAACAA/GT1/, Allele Primer 2/rhAmp-Y/ATACTTATTTACGCTTGAACCTCArAACAA/GT1/ and Locus Primer GCTCTGCAGGTATGTATTCATAGACTrCAAAA/GT2/.

## 3. Results

### 3.1. Sequence Coverage and Mutation Analysis

A total of 43 samples were analyzed after passing the quality controls for processing. This dataset provided a vast new reservoir of data. The exome sequencing specifications of the 43 DILI patient samples are shown in Table 1.

WES identified a total of 2,578,853 variants in the 43 cases of pediatric patient samples diagnosed with DILI. From these, those variants whose ClinVar classification term was pathogenic, probably pathogenic, drug responsive, or of uncertain significance were selected. In addition, those variants that had a synonymous effect were discarded. The total number was reduced to 4871, and from this set of variants, a protein–protein interaction and biological enrichment network was established, the result of which is shown in the Appendix A.

The analysis focused on the pathology of this study by excluding genes that were not associated with DILI according to the DISGENT and Phenopedia databases. In addition, so that the variants identified would have certain relevance in this study, only those with a frequency of more than 80% in our samples were selected.

After applying the depicted workflow (Figure 1), our analysis identified 361 variants (corresponding to the ABCB1, CYP2B6 and NAT2 genes) that could be interesting for this study. Each step is explained in detail in the following sections.

### 3.2. Validation of DILI-Related Variants

DILI-associated genes were obtained from the DisGeNET and Phenopedia databases. These genes were matched against the variants we found. In order to obtain a more robust comparison, those variants that appeared in more than 80% of our samples were selected. As a result, the ABCB1, NAT2 and CYP2B6 genes were obtained (Figure 2).

### 3.3. Functional Role of NAT2 Variants in DILI

#### 3.3.1. Prediction of Function

The computational tools described in the previous sections were used to determine whether these variants had a significant effect on the protein. For this purpose, the prediction scores were calculated. The thresholds used were those recommended by the literature for each tool. Those genes whose prediction was deleterious or probably damaging in at least one of the tools were kept (Table 2).

#### 3.3.2. Frequency of SNPs Linked to DILI and Statistical Analysis

Once the SNP analysis results were obtained from the 43 pediatric patient samples, they were matched against the 28 controls that were previously processed. For this purpose, the NAT2 gene was searched for in the sequencing of the controls, and the possible existence of the two variants selected for this study (rs1799930 and rs1801280) was verified.

The result in the case of the controls was that of the 28 samples sequenced by WES, only one of them had the rs1799930 variant and two of them had the rs1801280 variant (Figure 3).

Considering that the results indicated that 41 of the 43 samples from pediatric patients with DILI had at least one of the identified variants and that in the case of the controls, only 3 in 25 had at least one of these variants, the result of the statistical analysis of the observed cases was significant (*p* < 0.001; OR = 0.007) for the association of the pathology with having at least one of the two NAT2 variants.

#### 3.3.3. Frequency of Variants

The most frequent variants described within the NAT2 gene were analyzed in the cases and controls, obtaining the differential frequency between patients with DILI and healthy patients. Of the 13 SNPs compared, two variants (rs1799930 and rs1801280) stand out as being 12.7 and 6.9 times more frequent in patients with DILI, respectively (Table 3).

The deleterious variants of the NAT2 gene appeared in 41 of 43 samples: in 14 samples, we identified the c.341T>C mutation; in 12 samples, we identified the c.590G>A mutation; and in 15 samples, both appeared simultaneously.

The mean population frequency of all of the SNPs analyzed in the NAT2 gene and described as variants of interest in its activity were found to have mean levels of 7.9% in the controls and 14.8% in the cases, with significant differences noted for rs1799930 and rs1801280, where their frequency associated with DILI increased by 12.3 and 6.9 times, respectively.

#### 3.3.4. Real-Time Quantitative PCR Validation

Real-time PCR of the variants of interest was used to validate the results obtained by WES. 

The results shown in Figure 4 and Table 4 confirm the relationship of the genotypes for both SNPs in NAT2 with an increased risk of or predisposition to triggering a DILI process, regardless of whether these variants are found in homozygosis or heterozygosis.

Genotyping confirmed the results of significantly elevated frequency in the cases versus the controls, with 8 out of 10 cases being heterozygous forms versus homozygous forms, so that the presence of one or both alleles with one or both variants present (in 25% of the cases) are equally determinant as risk factors for DILI.

## 4. Discussion

The presented work reports for the first time the possible relationship of NAT2 polymorphisms with global susceptibility to DILI in the pediatric population. 

The advent of digital, electronic and molecular technologies has enabled the study of whole genomes. The integration of this information into drug development has opened the door to pharmacogenomic interventions in direct patient care. In the future, pharmacogenomics will make it possible to better identify the drug of choice and to optimize dosing regimens based on an individual’s genetic characteristics. 

The rs1799930 and rs1801280 polymorphisms, present in 41 of the 43 patients studied, constitute haplotypes associated with decreased enzyme activity (slow acetylation) due to Arg197Gln and Ile114Thr protein substitutions (NAT2*6 and NAT2*5, respectively) [13,18].

The two variants reported in this work for their implication in the possible development or triggering of DILI are known, previously described SNPs that involve an amino acid change and lead to a reduction in the acetylating activity of the enzyme. The aim of our work is to determine the causal relationship from a clinical point of view, which may open the door to deepening our understanding of the mechanism and function involved in these amino acids.

A prediction of the functional effect of amino acid substitution was carried out by the SIFT, PROVEAN, AlingGVGD, Panther and PolyPhen tools.

The resolved structure of wild-type human NAT2 (PDB 2PFR, https://doi.org/10.2210/pdb2PFR/pdb (accessed on 25 April 2024)) reveals the localization of both amino acids (Arg197 and Ile114) in locations that may be determinant in the functional structure and activity of the enzyme.

The amino acid changes are located at external sites of the protein structure, with possible implications in the half-life of the enzyme by favoring protein ubiquitination [19]. Reduced NAT2 activity has been associated with increased susceptibility to certain diseases, such as systemic lupus erythematosus, some types of cancer (hepatocellular carcinoma, bladder cancer, etc.) and increased toxicity to some drugs, such as isoniazid and hydralazine [20].

Prior to the present work, multiple studies have highlighted the importance of different NAT2 polymorphisms, especially the slow acetylator phenotype, in susceptibility to drug-induced hepatotoxicity, particularly in the context of anti-tuberculosis treatment. Specifically, two relevant systematic reviews and meta-analyses explored the relationship between NAT2 polymorphisms and drug-induced liver injury (DILI) in the context of tuberculosis treatment. The first study evaluated the potential association between NAT2 polymorphisms and DILI during anti-tuberculosis treatment. It included 37 studies with 1527 cases and 7184 controls, and found that the slow NAT2 acetylator phenotype was associated with an increased risk of DILI during anti-tuberculosis treatment. The analysis showed variability in risk among different ethnic populations, highlighting the importance of considering genetic diversity in DILI risk assessment [20,21].

The events triggered in the pediatric cases under study were idiosyncratic, and could not be related to any specific drug treatment, especially to those that have been described as particularly relevant for metabolization by NAT2, such as isoniazid, sulfamethoxazole, sulfasalazine, hydralazine, sulphadoxine, procainamide and dapsone [22]. The added Appendix A provides justification for this fact.

The second analysis included 14 studies with 474 cases and 1446 controls, and also confirmed a significant association between NAT2 slow acetylators and the risk of anti-tuberculosis drug-induced liver injury. This study suggests that patients with tuberculosis who are slow acetylators have an increased risk of DILI compared with other acetylation phenotypes. This finding underscores the potential utility of assessing NAT2 polymorphisms for the clinical prediction and prevention of DILI in patients undergoing anti-tuberculosis treatment [23,24].

The present study was performed with a broad group of drugs and was conducted in the pediatric population.

The presence of NAT2 polymorphisms, considered to be slow acetylators, has been associated not only with increased risk of DILI by anti-tuberculosis drugs, but also with increased severity of liver damage [25,26]. Likewise, the slow-acetylator phenotype has been associated with an increased risk of DILI by other drugs, such as trimethropyrim-sulfamethoxazole, in children and adults [27,28]. 

The analyses presented were performed with a broad group of drugs and were conducted in a pediatric population (see Appendix A).

As in adults, polymorphisms in the NAT2 gene can affect the metabolism of drugs and chemicals in children. However, the effects of NAT2 polymorphisms in children may be different from those in adults due to differences in physiology and the development of the enzyme system.

NAT2 activity in children may be different from that in adults, and may vary according to age and developmental stage. While the ontogeny of phase I enzymes is well established, age-dependent maturation changes in phase II enzymes such as NAT2 are not yet well understood. The maturation of NAT2 has been proposed to be achieved at around 5.3 years of age [29].

Human growth and physiological maturation are nonlinear processes, with discordant changes that have a direct effect on drug pharmacokinetics [28,29]. In fact, it has become evident that even in adults, the relationships between different pharmacokinetic parameters and covariates such as age and weight are not only nonlinear, but have high-order interactions, with data areas of discontinuity in the relationships [30,31,32,33].

In this regard, the importance of considering pharmacogenetics and enzymatic maturation in the design of therapeutic regimens for neonates, especially in low-birth-weight or preterm infants, to optimize the efficacy and safety of isoniazid treatment has been previously noted. Beranger et al. investigated the influence of the NAT2 genotype and maturation on isoniazid exposure in low-birth-weight and preterm infants with or without HIV exposure. This pharmacokinetic population analysis included 57 newborns with a mean gestational age of 34 weeks, and followed their progress for 6 months. It was observed that 90% of NAT2 maturation is reached at 4.4 months postnatally, and that the NAT2 genotype, along with weight and postmenstrual age, significantly influences isoniazid exposure. These results suggest that isoniazid dosing in low-birth-weight and preterm infants needs frequent adjustments to account for growth and maturation, and that pharmacogenetics could be key to establishing safe and effective dosing regimens [34].

Zhu et al. examined the role of the NAT2 genotype and enzyme maturation in isoniazid pharmacokinetics in perinatally HIV-exposed South African infants participating in a randomized controlled trial. Isoniazid plasma concentration measurements of 151 infants, starting at 3–4 months of age and receiving oral doses of 10 to 20 mg/kg/day for 24 months, were analyzed. The findings showed different enzyme maturation profiles for each of the three acetylation groups, with a significant increase in apparent isoniazid clearance in the fast and intermediate accelerators from 3 to 24 months of age, but no significant change in the slow-accelerator group. This suggests that the enzymatic maturation process of NAT2 is genotype-dependent [35]. 

From the above, the data presented suggest that a specific challenge in the implementation of pharmacogenomics in pediatrics is the result of maturation in pharmacogenomic phenotypes, such as variability in the maturation of drug-metabolizing enzymes and transporters, which may affect drug response. Ontogeny, or the impact of maturation on these enzymes and transporters, is an intrinsic factor in drug response that can complicate the implementation of pharmacogenomics in pediatric populations.

Our findings underscore the complexity and importance of NAT2 genetic polymorphisms in individual drug response and the risk of hepatotoxicity. Consideration of these genetic factors could be crucial for improving treatment outcomes in patients, particularly in those undergoing prolonged treatment with potentially hepatotoxic drugs.

## 5. Conclusions

In general, evaluating NAT2 polymorphisms in children through genetic testing could be used to personalize pharmacological treatment and assess the risk of medication toxicity. However, it is important to consider that interpreting genetic test results in children may differ from those in adults and may require a more careful and age-appropriate approach. As a result of this study, it can be concluded that the identified variants can be described as a risk factor for causing DILI in pediatrics.

## Figures and Tables

**Figure 1 biomedicines-12-01288-f001:**
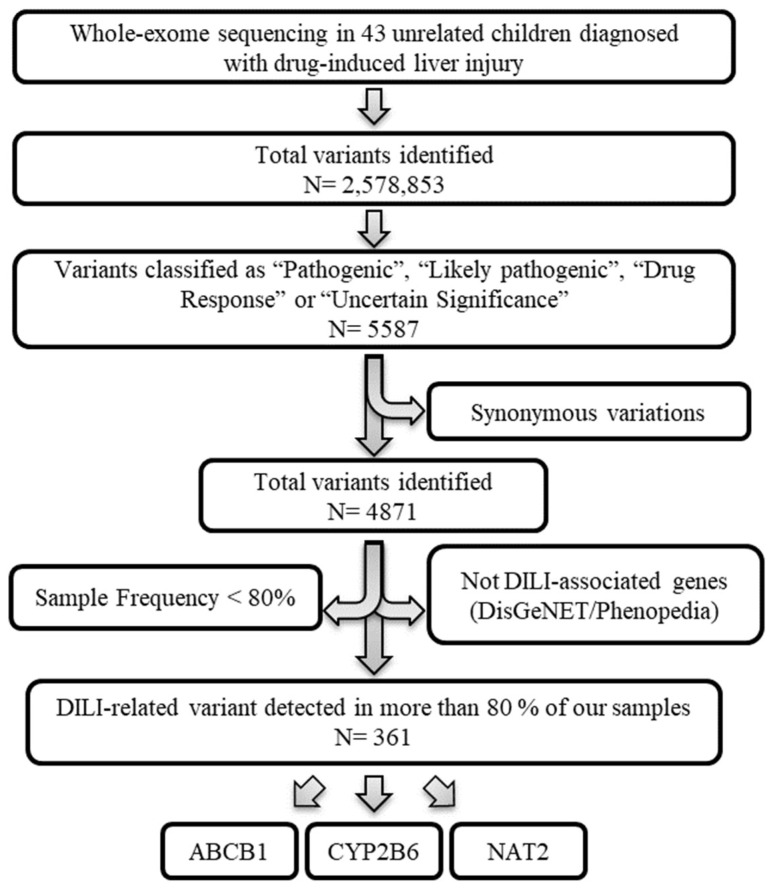
Variant selection workflow. Whole-exome sequencing data from 43 unrelated children diagnosed with drug-induced liver injury. Variants classified as “Pathogenic”, “Likely pathogenic”, “Drug Response” or “Uncertain Significance” by ClinVar were prioritized. The variations with synonymous effects were discarded. Next, those gene variants whose genes were not associated with DILI based on the DisGeNET (GDA-score > 0.3) and Phenopedia databases were excluded. Finally, only gene variants whose frequency in our samples was greater than 80% were used.

**Figure 2 biomedicines-12-01288-f002:**
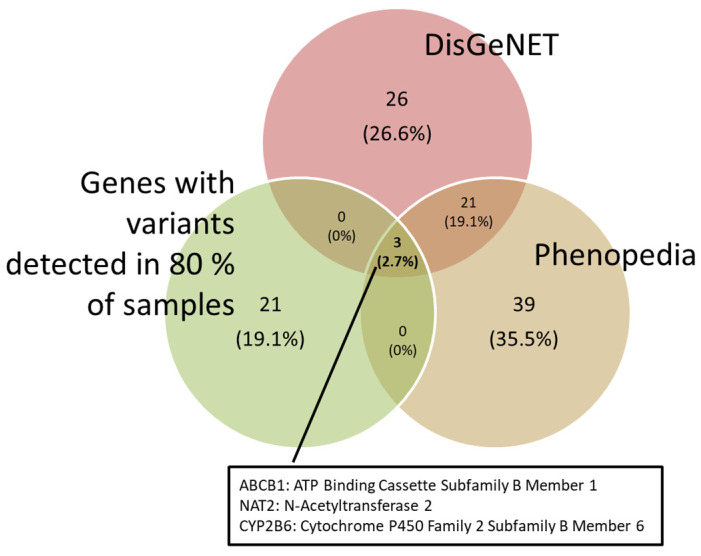
Venn diagram showing the genes associated with DILI from the DisGeNET and Phenopedia databases, and the genes with variations found in 80% of our samples.

**Figure 3 biomedicines-12-01288-f003:**
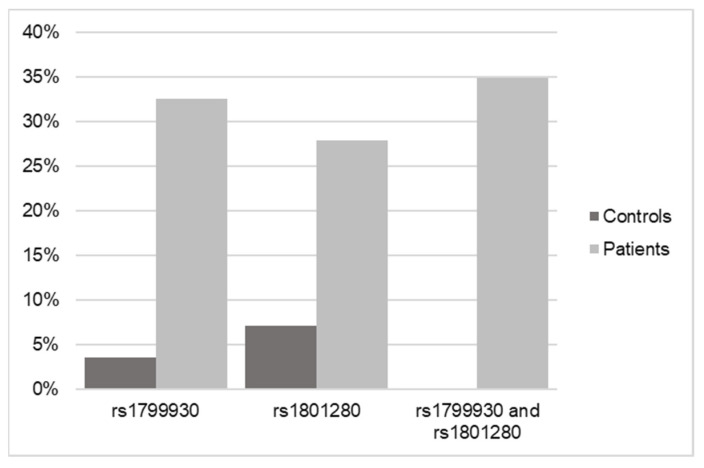
Frequency data of SNPs in NAT2 in control versus DILI patients according to WES.

**Figure 4 biomedicines-12-01288-f004:**
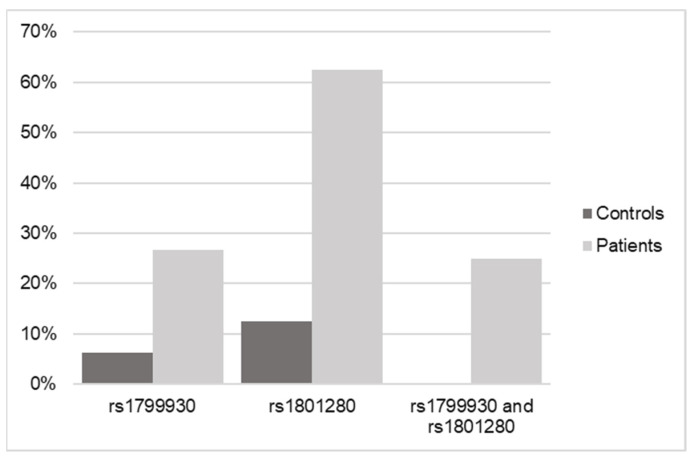
Frequency data of SNPs in NAT2 in control versus DILI patients according to real-time qPCR.

**Table 1 biomedicines-12-01288-t001:** Mean values in the exome sequencing data specifications of the 43 patients diagnosed with DILI.

Data Specification	Mean ± SD
Total reads	6.48 × 10^7^ ± 8.96 × 10^6^
Total yield (bp)	8.98 × 10^9^ ± 1.25 × 10^9^
Average read length (bp)	1.39 × 10^2^ ± 4.33 × 10^0^
Average throughput depth of target regions (X)	2.71 × 10^2^ ± 3.78 × 10^1^
Initial mappable reads (mapped to human genome)	6.48 × 10^7^ ± 8.96 × 10^6^
Non-redundant reads	5.31 × 10^7^ ± 7.94 × 10^6^
On-target reads	4.29 × 10^7^ ± 8.87 × 10^6^
Number of on-target genotypes (more than 1×	3.29 × 10^7^ ± 5.58 × 10^4^
Number of on-target genotypes (more than 10×)	3.26 × 10^7^ ± 3.26 × 10^5^
Number of on-target genotypes (more than 20×)	3.23 × 10^7^ ± 1.84 × 10^5^
Mean depth of target regions (X)	1.30 × 10^2^ ± 2.69 × 10^1^
Number of SNPs	6.00 × 10^4^ ± 1.11 × 10^3^
Number of synonymous SNPs	1.16 × 10^4^ ± 1.47 × 10^2^
Number of Missense Variants	1.09 × 10^4^ ± 1.84 × 10^2^
Number of Stop Losses	1.47 × 10^1^ ± 2.65 × 10^0^
Number of indels	8.78 × 10^3^ ± 5.20 × 10^2^
Number of Frameshift Variants	2.29 × 10^2^ ± 1.98 × 10^1^
Number of Inframe Insertions	1.86 × 10^2^ ± 8.32 × 10^0^
Number of Inframe Deletions	2.63 × 10^2^ ± 1.18 × 10^1^
% Found in dbSNP138	9.32 × 10^1^ ± 3.07 × 10^−1^
% Found in dbSNP154	9.31 × 10^1^ ± 4.06 × 10^−1^
Het/Hom ratio	1.69 × 10^0^ ± 1.19 × 10^−1^
Ts/Tv ratio	2.40 × 10^0^ ± 1.79 × 10^−15^

SD = standard deviation; bp = base pair; SNP = single-nucleotide polymorphism; Het/Hom = heterozygous/homozygous; Ts/Tv = transition/transversion.

**Table 2 biomedicines-12-01288-t002:** Prediction data for NAT2 variants.

Feature	Variant 1	Variant 2
Gene symbol	NAT2	
Description	N-Acetyltransferase 2	
rsID	rs1799930	rs1801280
Variation name	NM_000015.3	NM_000015.2
HGVS.c	c.590G>A	c.341T>C
HGVS.p	p.Arg197Gln	p.Ile114Thr
SIFT score/pred	0.054/T ^1^	0.048/D ^2^
PROVEAN score/pred	−2.84/D ^2^	−4.18/D ^2^
Align GVGD	42.81 (Class C35) ^3^	89.28 (Class C65) ^3^
Panther	Probably damaging	Possibly damaging
Polyphen 2	Probably damaging	Possibly damaging

^1^ Tolerable; ^2^ deleterious; ^3^ Class C35/Class C65: probability of interfering with protein function is medium/high, respectively.

**Table 3 biomedicines-12-01288-t003:** NAT2 gene variant describes by WES as highly frequent located in samples from controls and DILI patients.

	Controls		Patients	
NAT2 SNP	Proportions	%	Proportions	%
rs1801280	2	7.1	21	48.8
rs1799930	1	3.6	19	44.2
rs1801279	6	21.4	8	18.6
rs4986997	5	17.9	8	18.6
rs1041983	3	10.7	6	14.0
rs1565684	3	10.7	5	11.6
rs1799929	2	7.1	5	11.6
rs1208	1	3.6	2	4.7
rs1799931	1	3.6	2	4.7
rs4271002	2	7.1	2	4.7
rs4345600	1	3.6	2	4.7
rs9987109	1	3.6	2	4.7
rs4986996	1	3.6	1	2.3

**Table 4 biomedicines-12-01288-t004:** Genotiping frecuency of NAT2 variants in controls and DILI patients.

	n	SNP1 Count	SNP1 %	SNP2 Count	SNP2 %	SNP1-2 Count	SNP1-2 %
Controls	16	1	6.25	2	12.50	0	0
Patients	8	3	37.50	5	62.50	2	25.00

n = sample size; SNP1 = rs1799930; SNP2 = rs1801280.

## Data Availability

The analyzed data of the present study are available to any researcher upon request to any of the authors.

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
