# Peer review of "N-acetyltransferase Gene Variants Involved in Pediatric Idiosyncratic Drug-Induced Liver Injury"

_biomedicines, 2024, doi:10.3390/biomedicines12061288_

Round 1

Reviewer 1 Report

Comments and Suggestions for Authors

Alés-Palmer et al performed WES on 43 pediatric patients diagnosed with DILI and identified some important variants associated with this pathology. It is an interesting study. However, some experiments still need to confirm their finding.

1. All pathogenic mutations must be verified by standard sequencing as Sanger, such as p.Arg197Gln, p.Ile114Thr and some potential Rs pathogenic

2.  To Predicting the functional effect of amino acid substitutions, it needs to further perform in silico prediction

3. Since many genes have been found in the WES, it would be great to perform network gene/protein analysis such as CluGo, String..

4. Frequency of variants, the authors need to discuss their findings, how each SNP could affect the case, and compare the control...

Comments on the Quality of English Language

Major revision needs

Author Response

Response to Reviewer 1 and Editor

Manuscript ID: biomedicines-2978099

First, we want to thank for your effort in reviewing our manuscript titled “NAT2 gene variants involved in pediatric idiosyncratic drug-induced liver injury” and for your constructive comments, which have undoubtedly contributed to improve the quality of our manuscript.

Please, find below the responses to your kind suggestions:

Comment:

Alés-Palmer et al performed WES on 43 pediatric patients diagnosed with DILI and identified some important variants associated with this pathology. It is an interesting study. However, some experiments still need to confirm their finding.

  1. All pathogenic mutations must be verified by standard sequencing as Sanger, such as p.Arg197Gln, p.Ile114Thr and some potential Rs pathogenic

Reply:

The risk variants described as risk factors for DILI are SNPs that have been found by high depth WES, and confirmed in the samples by genotyping using specific probes designed and validated by RT-PCR for the variants present in the NAT2 gene. Therefore, we consider that the comment referred to by the reviewer has already been experimentally addressed in the work by a specific genotyping method on the specific variants using fluorescence channels that support the sequencing results obtained in patients versus controls.

Comment:

  1. To Predicting the functional effect of amino acid substitutions, it needs to further perform in silico prediction

Reply:

The 2 variants reported in this work for their implication in the possible development or triggering of DILI are known SNPs previously described that involve an amino acid change and lead to a reduction in the acetylating activity of the enzyme. The aim of our work is the causal relationship from a clinical point of view, which may open the door to deepen the mechanism and function involved in these amino acids.

Prediction of functional effect of amino acid substitution has been carried out by SIFT and PROVEAN tools, and we can complete with Align GVGD, Panther and PolyPhen. We included this comments into the text.

The resolved structure of wild-type human NAT2 (PDB 2PFR, https://doi.org/10.2210/pdb2PFR/pdb) reveals a localization of both amino acids (Arg197 and Ile114) in locations that may be determinant in the functional structure and activity of the enzyme.

The amino acid changes are located in external sites of the protein structure, with possible implication on the half-life of the enzyme by favoring protein ubiquitinization. [Wu H, Dombrovsky L, Tempel W, Martin F, Loppnau P, Goodfellow GH, Grant DM, Plotnikov AN. Structural basis of substrate-binding specificity of human arylamine N-acetyltransferases. J Biol Chem. 2007 Oct 12;282(41):30189-97. doi: 10.1074/jbc.M704138200. Epub 2007 Jul 26. PMID: 17656365].

Once variants with a high frequency in our samples that were related to DILI were obtained, computational tools such as SIFT (https://www.ncbi.nlm.nih.gov/pmc/articles/PMC3394338/), PROVEAN (https://www.ncbi.nlm.nih.gov/pmc/articles/PMC4528627/), Align GVGD (https://www.ncbi.nlm.nih.gov/pmc/articles/PMC2563222/), Panther (https://pubmed.ncbi.nlm.nih.gov/27193693/) and PolyPhen 2(https://pubmed.ncbi.nlm.nih.gov/23315928/) were used to determine which of these variants might be causative of the pathology or characteristic phenotype in DILI. Default thresholds were used to determine whether it was a deleterious/pathogenic mutation based on the prediction of each tool.

The computational tools described in the previous sections were used to determine whether these variants had a significant effect on the protein. For this purpose, the prediction scores were calculated. The thresholds used were those recommended by the literature for each tool. Those genes whose prediction was deleterious or probably damaging in at least one of the tools were kept (Table 2).

Feature

Variant 1

Variant 2

Gene Symbol

NAT2

Description

N-Acetyltransferase 2

rsID

rs1799930

rs1801280

Variation Name

NM_000015.3

NM_000015.2

HGVS.c

c.590G>A

c.341T>C

HGVS.p

p.Arg197Gln

p.Ile114Thr

SIFT score/pred

0.054/T1

0.048/D2

PROVEAN score/pred

-2.84/D2

-4.18/D2

Align GVGD

42.81 (Class C35)3

89.28 (Class C65)3

Panther

Probably damaging

Possibly damaging

Polyphen 2

Probably damaging

Possibly damaging

1Tolerable; 2Deleterious; 3Class C35/Class C65: Probability of interfering with protein function medium/high, respectively.

Comment:

  1. Since many genes have been found in the WES, it would be great to perform network gene/protein analysis such as CluGo, String..

Reply:

We have performed a protein-protein interaction network combined with enrichment by gene ontology and others, which we attach in the supplementary material (Appendix A).

Comment:

  1. Frequency of variants, the authors need to discuss their findings, how each SNP could affect the case, and compare the control...

Reply:

The mean population frequency of any of the SNPs analyzed in the NAT2 gene and described as variants of interest in its activity, have been found at mean levels of 7.9% in controls and 14.8% in cases, the significant difference being for rs1799930 and rs1801280, where their frequency associated with DILI increases 12.3 and 6.9 times respectively.

Genotyping confirmed the results of significantly elevated frequency in cases versus controls, with 8 out of 10 cases being heterozygous forms versus homozygous forms, so that the presence of one or both alleles with one or both variants present (in 25% of the cases), are equally determinant as a risk factor for DILI.

We sincerely appreciate your valuable comments and suggestions and hope that our responses adequately address your concerns.

Kind regards,

The authors

Reviewer 2 Report

Comments and Suggestions for Authors

The authors sequenced the exome of 43 pediatric patients diagnosed with idiosyncratic drug-induced liver injury and discovered the presence of two variations in the N-acetyltransferase gene 2 in a significant number of patients, while these variations were virtually absent in controls, allowing the authors to propose these variations as a risk factor for this idiosyncratic drug-induced liver injury. The results are relevant and highlight the need to consider genetic factors in patients undergoing chronic treatment with potentially hepatotoxic drugs.

The work is well done and the results are robust. I have no major objections. However, I would like to suggest some minor points to the authors for their consideration.

I understand the difficulties in obtaining biological samples from healthy pediatric individuals and this is the reason for using adult controls. This should not be relevant since the differences between children and adults should be at the transcriptomic level and not at the exome level. But keeping this in mind, why not do this work with adult patients? It should be easier to get a larger number of samples and the results would be extrapolable to pediatrics.

Did the authors try to associate each NAT2 variant with specific drugs listed in the supplementary table? The number of samples is relatively small and there is probably not enough statistical power, but this information could be interesting.

It is typical in other omic papers to publish the raw data in open free repositories. Did the authors consider this possibility? It would allow readers to reanalyze the information and would help the paper to meet the principles of open science.

I think the authors should add some more information about the software used for the bioinformatic analysis. Is it open, publicly available software? If so, please provide the source and a link to access it. Please also add the version or any other information you consider relevant in this regard.

MINOR POINTS

Table 3 and Figure 3, as well as Table 4 and Figure 4, actually duplicate the same information. Perhaps this is an editorial issue to decide whether such publication is convenient or not. From this reviewer's point of view, Figure 3 is more illustrative than Table 3 and Table 4 is more illustrative than Figure 4.

For the sake of clarity for the readers, I suggest adding the URL addresses of the databases used in this paper.

Author Response

Response to Reviewer 2 and Editor

Manuscript ID: biomedicines-2978099

First, we want to thank for your effort in reviewing our manuscript titled “NAT2 gene variants involved in pediatric idiosyncratic drug-induced liver injury” and for your constructive comments, which have undoubtedly contributed to improve the quality of our manuscript.

Please, find below the responses to your kind suggestions:

Comment:

The authors sequenced the exome of 43 pediatric patients diagnosed with idiosyncratic drug-induced liver injury and discovered the presence of two variations in the N-acetyltransferase gene 2 in a significant number of patients, while these variations were virtually absent in controls, allowing the authors to propose these variations as a risk factor for this idiosyncratic drug-induced liver injury. The results are relevant and highlight the need to consider genetic factors in patients undergoing chronic treatment with potentially hepatotoxic drugs.

The work is well done and the results are robust. I have no major objections. However, I would like to suggest some minor points to the authors for their consideration.

I understand the difficulties in obtaining biological samples from healthy pediatric individuals and this is the reason for using adult controls. This should not be relevant since the differences between children and adults should be at the transcriptomic level and not at the exome level. But keeping this in mind, why not do this work with adult patients? It should be easier to get a larger number of samples and the results would be extrapolable to pediatrics.

Reply:

The triggering and risk processes of DILI processes in pediatric patients are different from those in adults, so the extrapolation of study results between populations can be confusing. In any case, we consider that the proposed transcriptomic study of the NAT2 pathway in adult and pediatric cohorts with and without DILI could be an interesting way to confirm this casuistry.

Comment:

Did the authors try to associate each NAT2 variant with specific drugs listed in the supplementary table? The number of samples is relatively small and there is probably not enough statistical power, but this information could be interesting.

Reply:

The events triggered in the pediatric cases under study were idiosyncratic, and could not be related to any specific drug treatment, especially to those that have been described as particularly relevant for metabolization by NAT2, such as Isoniazid, sulfamethoxazole, sulfasalazine, hydralazine, sulphadoxine, procainamide, dapsone (McDonagh et al., Pharm. Gen. 2014; PhamGKB Summary: Very Important Pharmacogene information for N-acetyltransferase 2). The added supplementary table provides justification for this fact.

Comment:

It is typical in other omic papers to publish the raw data in open free repositories. Did the authors consider this possibility? It would allow readers to reanalyze the information and would help the paper to meet the principles of open science.

Reply:

We consider this possibility in order to share the raw data, once completed and extracted all the publishable results of interest to the research group, so that they can also be of interest to other researchers.

Comment:

I think the authors should add some more information about the software used for the bioinformatic analysis. Is it open, publicly available software? If so, please provide the source and a link to access it. Please also add the version or any other information you consider relevant in this regard.

Reply:

We include this information by adding the requested details in the material and methods section of the work.

Comment:

MINOR POINTS

Table 3 and Figure 3, as well as Table 4 and Figure 4, actually duplicate the same information. Perhaps this is an editorial issue to decide whether such publication is convenient or not. From this reviewer's point of view, Figure 3 is more illustrative than Table 3 and Table 4 is more illustrative than Figure 4.

Reply:

Fully agreeing with your assessment, we eliminated Table 3 and Figure 4.

Comment:

For the sake of clarity for the readers, I suggest adding the URL addresses of the databases used in this paper.

Reply:

In full agreement with your assessment, we add the links to the databases of interest in their corresponding section.

We sincerely appreciate your valuable comments and suggestions and hope that our responses adequately address your concerns.

Kind regards,

The authors

Round 2

Reviewer 1 Report

Comments and Suggestions for Authors

Thank you to the author for responding to my remarks. Although these pathogenic variations have already been published, the authors still need to verify them based on their results by Sanger Seq, which is a standard experiment performed after WES. Sanger Seq is a common approach to use.

Round 3

Reviewer 1 Report

Comments and Suggestions for Authors

Thanks for the additional response. However, the electropherogram of Sanger sequencing profile obtained using WES (take through IGV software) must be provided for further verification.
